# Exploring Topological Semi-Metals for Interconnects

**Satwik Kundu** [1,*,†], **Rupshali Roy** [1,*,†], **M. Saifur Rahman** [1], **Suryansh Upadhyay** [1], **Rasit Onur Topaloglu** [2], **Suzanne E. Mohney** [1,3], **Shengxi Huang** [4] and **Swaroop Ghosh** [1,*]

1   School of Electrical Engineering and Computer Science, The Pennsylvania State University, Universty Park, PA 16801, USA
2   IBM Corporation, Armonk, NY 12533, USA
3   Department of Material Science and Engineering, The Pennsylvania State University, University Park, PA 16801, USA
4   Department of Electrical and Computer Engineering, Rice University, Houston, TX 77005, USA
*   Correspondence: sxk6259@psu.edu (S.K.); rzr5509@psu.edu (R.R.); szg212@psu.edu (S.G.)
†   These authors contributed equally to this work.

**Abstract:** The size of transistors has drastically reduced over the years. Interconnects have likewise also been scaled down. Today, conventional copper (Cu)-based interconnects face a significant impediment to further scaling since their electrical conductivity decreases at smaller dimensions, which also worsens the signal delay and energy consumption. As a result, alternative scalable materials such as semi-metals and 2D materials were being investigated as potential Cu replacements. In this paper, we experimentally showed that CoPt can provide better resistivity than Cu at thin dimensions and proposed hybrid poly-Si with a CoPt coating for local routing in standard cells for compactness. We evaluated the performance gain for DRAM/eDRAM, and area vs. performance trade-off for D-Flip-Flop (DFF) using hybrid poly-Si with a thin film of CoPt. We gained up to a 3-fold reduction in delay and a 15.6% reduction in cell area with the proposed hybrid interconnect. We also studied the system-level interconnect design using NbAs, a topological semi-metal with high electron mobility at the nanoscale, and demonstrated its advantages over Cu in terms of resistivity, propagation delay, and slew rate. Our simulations revealed that NbAs could reduce the propagation delay by up to 35.88%. We further evaluated the potential system-level performance gain for NbAs-based interconnects in cache memories and observed an instructions per cycle (IPC) improvement of up to 23.8%.

**Keywords:** local interconnect; global interconnect; Weyl semi-metal; propagation delay; instructions per cycle; cache design

## 1. Introduction

Conventional interconnect and via technologies are facing scalability challenges and pose limitations for density critical circuit components such as static RAM (SRAM) and peripherals such as wordline driver and sense amplifier. Some of these challenges are as follows:

**Local interconnect:** The high resistivity of lower-level metals such as M0-2 limit their scaling and consequently that of the standard cells footprint and SRAM bit-cell size with the technology node. Scaling the metal pitch to pack more tracks runs the risk of post-manufacturing shorts. Therefore, the benefit of transistor scaling is overshadowed by poor interconnect scalability. The high resistivity also confines the use of lower-level metals to very short distances. These challenges are slightly alleviated for intermediate metals, e.g., M3-5; nevertheless, packing more tracks remains a challenge to ensure the reliable delivery of global signals, e.g., power, clock, and miscellaneous DC signals to lower metals.

**Global interconnect:** Higher-level metals that carry global signals are highly capacitive due to their dimension consuming switching power and requiring buffers to boost the

signal strength. Techniques are needed to scale their dimension or to explore alternative signaling techniques to control the switching power and delay.

**Via and contact resistance:** The lower level vias including contacts with the substrate are highly resistive due to a tighter footprint. Placing multiple vias affects the density-sensitive elements, e.g., SRAM, negatively. Increasing the via dimension may lead to post-manufacturing defects due to tighter via-to-via spacing. Low resistivity/dissipation-less materials are essential to remove the above bottlenecks.

Topological semi-metals, such as WSMs, hold great promise as interconnects due to their unconventional resistivity scaling with reduced dimension. Thus, we evaluate the unique properties of Weyl semi-metals (WSMs) for applications as global and local interconnect to address some of the aforementioned challenges. We provide a background on Weyl semi-metals below.

**Weyl semi-metal:** A new topological phase of matter, the Weyl semi-metals (WSMs), has recently been uncovered. WSMs exhibit an electronic structure governed by linear band dispersions and degenerate (Weyl) nodes that lead to exotic physical phenomena such as chiral anomalies [1]. In WSMs, conduction and valence bands touch at Weyl nodes, forming quasiparticles that emerge as Weyl fermions and have definite chiralities that are considered a topologically protected chiral charge. A Fermi arc connects two Weyl nodes of opposite chirality only through the crystal boundary, where they are separated in momentum space by breaking either time-reversal symmetry or inversion symmetry. Since they are magnetic monopoles with momentum-space configurations, their spins are locked to their momentum directions [2]. These give WSMs topologically protected surface electronic states whose transport is backscattering-free and high-mobility.

The discovery of the Weyl semi-metal had to wait until recent advancements despite the various versions of Weyl semi-metal theory which have existed for a long time. This is due to the fact that accurate simulation and material characterization are necessary for experimental realization. Extensive efforts have been made to focus on time reversal-breaking materials as candidates for Weyl semi-metals [3]. However, it became clear that the "time-reversal breaking" approach of gathering suitable materials has a number of obstacles either demonstrating Fermi arcs or isolating the Weyl quasiparticles because of the strong correlations in magnetic materials and the destruction of sample quality upon magnetic doping. The search for Weyl semi-metal candidates in naturally occurring inversion-breaking non-centrosymmetric single crystals can, however, avoid the difficulties described above.

The inorganic crystal structure database of FIZ Karlsruhe [4] records the lattice structure of crystals that have been synthesized over the course of a century. By following this approach and calculating the band structure of materials that are likely to be semi-metals, many experimentally feasible candidate materials have been identified [5,6]. The TaAs class of compounds known as type I Weyl semi-metals, including NbAs and TaP, have already been experimentally realized [7]. Although it remains elusive about type II Weyl semi-metals that violate the Lorentz symmetry, few crystalline solids such as lanthanum aluminum germanide (LaAlGe) and $Ta_3S_2$ [5,6] can help in the observation of type II Weyl fermions. In order to investigate the salient feature of Weyl semi-metals, most effort has been performed on focused-ion beam (FIB) milling [8] to microstructure bulk crystals. Figure 1 shows how FIB works to fabricate the microstructure from bulk crystals.

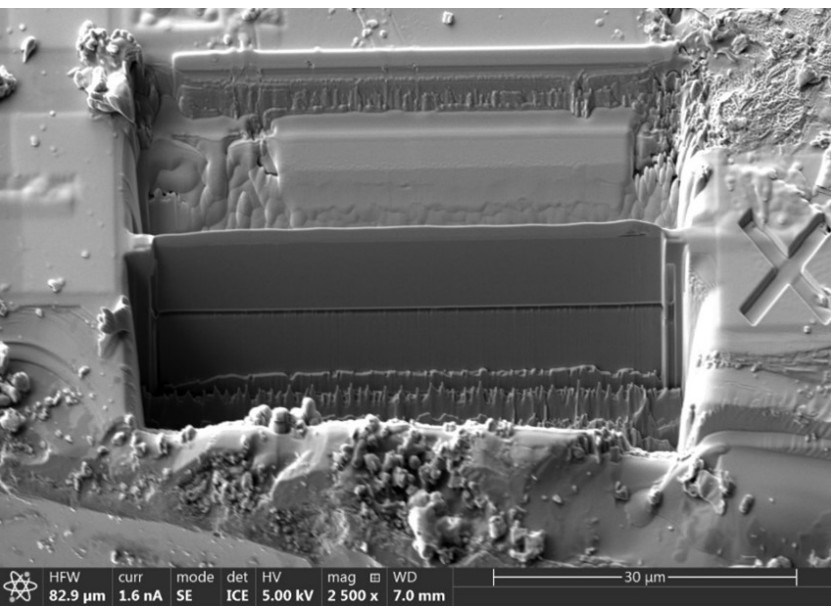

**Figure 1.** SEM micrograph of a fabricated microstructure from bulk crystal using FIB.

Despite these advancements, the study of WSMs is still in the early stage and requires concerted efforts between physicists, material scientists, and electrical engineers to achieve their successful application in integrated circuits. For example, the fundamental physical properties, such as the role of Fermi arc and Weyl nodes in the WSM conductivity, and their change with dimensionality, require in-depth investigations. The material synthesis techniques, especially for thin film and nanowires, are underdeveloped and require the attention of material scientists. The nano-fabrication recipes of the synthesized WSMs into devices and their heterogeneous integration into integrated circuits are worth further exploration by electrical engineers. In addition, the field will benefit from the high-throughput calculation to discover new WSMs that meet the requirement of performance and are compatible with semiconductor fabrication processes, which could be implemented by machine learning combined with first-principles calculations and process/circuit modeling.

In this paper, we experimentally demonstrated the improved resistivity scaling of CoPt over Cu at thinner dimensions (i.e., thickness below 10 nm). Since interconnects are typically not thinner than 10 nm, CoPt cannot be used to replace existing Cu-based interconnects. Therefore, we propose the deposition of CoPt on regular poly interconnect to develop a hybrid poly-WSM interconnect for local routing in standard cells and memory circuits for compactness. We evaluated the performance gain for DRAM/eDRAM and area vs. performance trade-off for DFF using a hybrid poly-WSM interconnect. We further investigated another topological semi-metal, NbAs, and demonstrated that it outperforms Cu as a global interconnect using various metrics across different process technologies. We also ran extensive simulations using the gem5 simulator to demonstrate the system/architecture level performance improvement provided by NbAs over Cu.

The rest of the paper is structured as follows: we discuss the fabrication of the CoPt interconnect and the evaluation methodology used for both lower and higher-level interconnect performance evaluation in Section 2, present the setup used for our analysis, and discuss the results in Section 3, and conclude in Section 4.

## 2. Methodology

In this section, we first present the CoPt fabrication methodology and hybrid poly-WSM interconnect.

Then, we discuss the application of hybrid poly-WSM as a local interconnect and evaluate its performance. Finally, we present another topological semi-metal, NbAs, which

is superior to Cu in terms of resistivity, and evaluate its performance at the circuit and system levels for global interconnect applications.

### 2.1. CoPt Fabrication

We deposited thin films of CoPt with different thicknesses onto $SiO_2/Si$ substrate by radio frequency (RF) magnetron sputtering in a background vacuum greater than $4 \times 10^{-7}$ Torr. The sputter deposition was carried out in an argon environment at a pressure of 5 mtorr at room temperature. We tuned the RF power supplies for each of the Co and Pt targets to achieve a stoichiometry close to 1:1. The exact composition ratio of the CoPt thin films was $Co_{48}Pt_{52}$, as confirmed by the energy dispersive X-ray spectroscopy (EDS) shown in Figure 2.

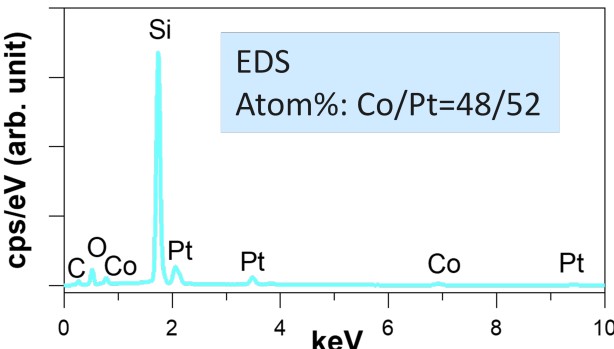

**Figure 2.** EDS spectrum of CoPt film to confirm the Co/Pt ratio is roughly 50/50.

For Co and Pt targets, the power was of 200 W and 39 W, respectively. After obtaining the correct stoichiometry, the next objective involved achieving the correct phase/crystalline structure. CoPt with stoichiometry Co:Pt = 1:1 has two phases: tetragonal and trigonal. Our target CoPt phase is tetragonal. Therefore, we annealed in $N_2$ our CoPt thin films at 700 degrees Celsius to have a tetragonal phase which was verified by XRD displayed in Figure 3. SEM images for the sample of CoPt with a thickness of 40 nm are shown in Figure 4a,b. The grain size is observed by SEM. Figure 4a indicates the CoPt film before annealing with a grain size of approximately 10 nm, and Figure 4b corresponds to the CoPt film after annealing with a grain size of 100 nm–1 μm. We then measured the film resistivity at different thicknesses, as shown in Figure 5b. At a film thickness of ≤10 nm, CoPt (22.2 ± 2.9 μΩ·cm), resistivity is lower than Cu (~36 μΩ· cm).

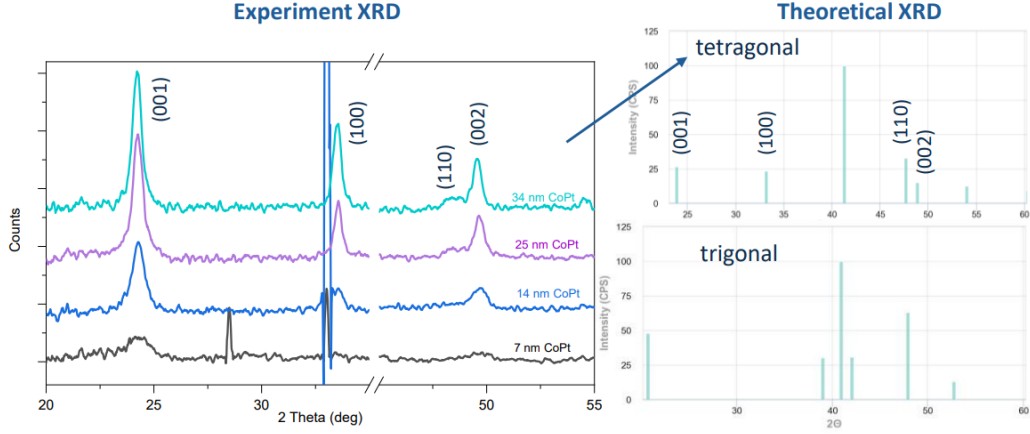

**Figure 3.** XRD of CoPt after 700 °C annealing in N2 atmosphere.

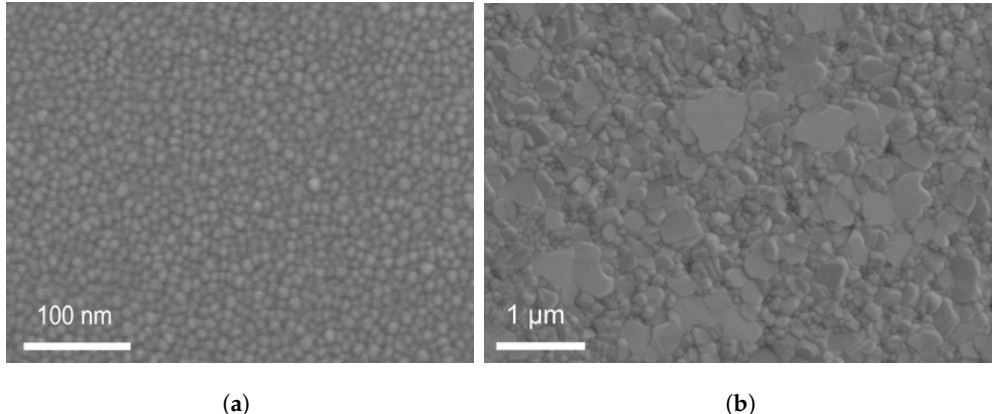

(**a**)            (**b**)

**Figure 4.** SEM images for the CoPt thin film (**a**) before annealing and (**b**) after annealing.

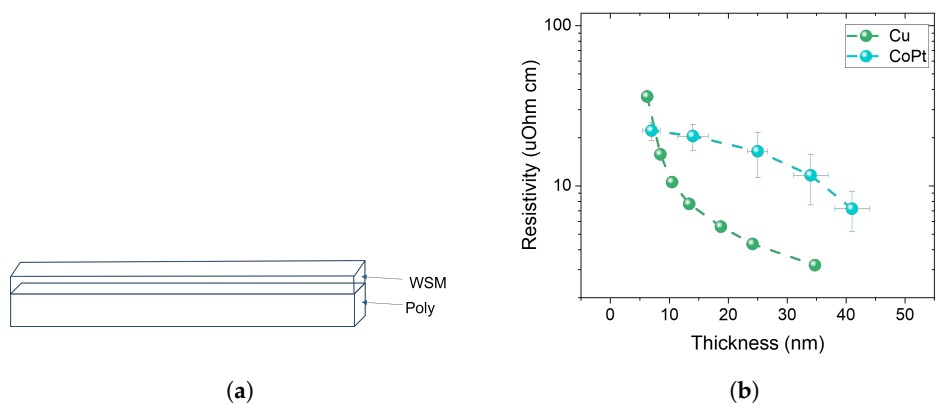

(**a**)            (**b**)

**Figure 5.** (**a**) Schematic for the deposition of CoPt on poly to create a hybrid poly-WSM interconnect. (**b**) The plot on the right shows how the resistivity of CoPt varies in thickness in comparison to Cu. At thickness < 10 nm, CoPt shows advantage over Cu.

### 2.2. Low-Level Interconnect

For low-level interconnect, we proposed hybrid poly-WSM (poly routing with a layer of CoPt deposited on it) and evaluate its resistance. Following that, we estimate various performance metrics such as propagation delay when used in various circuits. We also obtain area benefits due to the elimination of metal-1 and vias from the circuits.

### 2.3. High-Level Interconnect

Global interconnects, unlike local interconnects, cannot afford high resistivity because they travel over longer distances. For this study, we consider another WSM, i.e., NbAs-based material, to design high-level interconnects for memories and evaluate its impact on system performance. We used the gem5 simulator [9,10] to run extensive simulations to determine the advantage that NbAs could provide over Cu when used as a global chip interconnect. Gem5 offers a large number of configurable parameters and output metrics for evaluating the performance of a specific CPU configuration. However, for this work, we varied a subset of those parameters to determine the probable performance gain provided by NbAs interconnects.

Interconnect wire delay dominates cache latency [11]. As a result, the actual time taken to read or write a cell is usually just one cycle.

Since NbAs has higher conductivity and as a result, lower propagation delay compared to Cu at the nanoscale, we change the latencies of both the L1 and L2 (last level) cache accordingly to analyze the effect on CPU performance. L1 cache latency usually varies between 2 and 4 cycles depending on the cache size, whereas L2 cache latency usually varies between a few tens of cycles. For instance, the L1 and L2 cache latency for the P-cores

in Apple M1 chip [12], which is based on ARM's big.LITTLE architecture, comprising 3 cycles and 18 cycles, respectively.

For each cache level, gem5 allows users to modify three types of latencies: tag, data, and response latency. To measure the performance improvement, we altered all three latencies equally and monitored the impact on the system's instructions per cycle (IPC), which is the average number of instructions executed per clock cycle. A higher IPC generally indicates better system performance.

## 3. Evaluation

In this section, we present the circuit and system setup we used for evaluation and discuss our findings.

### 3.1. Setup

**NbAs-based interconnect:** A simple transmitter–receiver test circuit is used to carry out the simulations to study the behavior of the Cu and NbAs nanoribbons, respectively. A pulse of amplitude 1 V, initial delay 1 ns, rise and fall time of 1 ps each, a period of 2 ns, and duty cycle of 50% is given as input to a CMOS transmitter inverter. The output of this inverter is then passed through a nanoribbon of Cu/NbAs. At constant intervals, buffers are placed in the path of the pulse. The size of these buffers is 4-fold the size of the transmitter inverter. At the end of the ribbon, the output pulse is given as input to the receiver inverter. The size of the receiver varied in one of our experiments. We used 65 nm, 45 nm, and 22 nm predictive technology models (PTM) [13] for simulations. The values for resistivity for the Cu and NbAs materials were taken from the plot in Figure 6 [14].

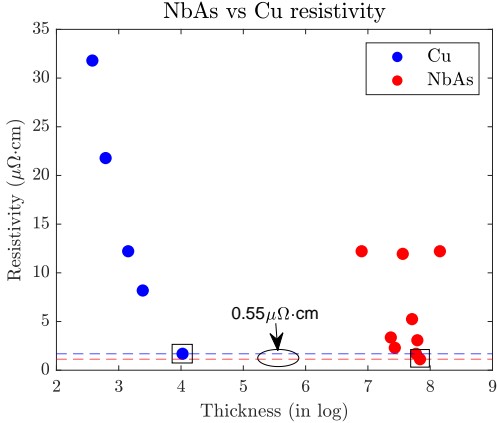

**Figure 6.** Cu and NbAs resistivity values at the nanoscale dimension [14]. We plotted resistivity against log thickness (originally in nm) values for easier interpretation. We also indicated the Cu and NbAs resistivity values we used in our analysis (with square). The NbAs resistivity is 0.55 μΩ·cm or ≈32% lower than Cu.

**GEM5:** GEM5 simulator provides a diverse set of CPU models, instruction set architectures (ISAs), memory systems, etc., which help in conducting computer architecture research. GEM5 supports two main system modes and four different CPU models [15]. For our experimentation, we used a system called emulation mode (SE) and the out-of-order CPU model.

The gem5 simulator has modular support for multiple ISAs (ARM, RISC-V, SPARC, etc.). We ran our simulations on the X86 architecture. All of the experiments were performed using the gem5 detailed processor, *DerivO3CPU* clocked at 2 GHz. Table 1 shows the full gem5 system configuration used for simulations.

**Table 1.** GEM5 configuration.

| Parameter | Value |
|:---:|:---:|
| ISA | x86 |
| CPU | DerivO3CPU |
| CPU model | Out-of-order |
| Core frequency | 2 GHz |
| Cores | 1 |
| L1-I size | 64 kB |
| L1-D size | 128 kB |
| L1 associativity | 2 |
| L1 latency | 4 |
| L2 size | 2048 kB |
| L2 associativity | 8 |
| L2 latency | 15 |
| Cacheline size | 64 B |

Gem5 supports a number of benchmark suites such as SPEC CPU 2017, SPLASH-2, NPB (NAS Parallel Benchmarks), etc. We used the Parsec 3.0 benchmark suite [16] for our evaluation purpose. It includes a wide spectrum of emerging applications in recognition, mining, and synthesis (RMS) as well as systems applications that mimic larger-scale multithreaded commercial programs. Out of the available 13 main benchmarks, we used 5 of them for our experiments, namely *Blackscholes*, which analytically calculates the prices for a portfolio of European options with the Black–Scholes partial differential equation; *Canneal*, which uses simulated cache-aware annealing to optimize the routing cost of a chip design; *Fluidanimate*, uses fluid dynamics for animation purposes with smoothed particle hydrodynamics; *Raytrace*, real-time tracing and *Streamcluster*, solves the online clustering of an input system.

Additionally, we also used another CPU benchmark released by [17] to evaluate the system performance on deep learning workloads. The benchmark involves testing a simple MLP with three hidden layers, 1024 neurons in the first two layers and 26 neurons (number of classes) in the final layer after it was trained on a synthetic dataset. All the simulations were performed on Gem5 version 20.1.0 on a Ubuntu system with Intel(R) Core(TM) i9-10900X CPU @ 3.70 GHz and 16 GB Ram.

### 3.2. Results

**Hybrid poly-WSM interconnect:** For the evaluation of the hybrid poly-WSM interconnect, we considered two test cases, namely:

*(a) DRAM/eDRAM:* The wordline of the DRAM/eDRAM designs is typically routed using poly-Si to achieve the best memory density. However, it severely degrades the wordline (WL) performance due to high poly-Si resistivity. To recover the performance, the WLs are also routed in metal-3 and the poly-Si WLs are occasionally shorted with metal-3 (called strapping). Since metal-3 is much faster than poly-si, the worst-case WL delay is alleviated with this approach. However, the contact with poly-Si degrades the array density. By tuning the number of metal-3 and poly-Si connection/straps, one can make a trade-off between performance and memory density. To evaluate the DRAM/eDRAM performance with the hybrid poly-WSM interconnect, we assume three baseline cases of 256-bit DRAM designs. In Figure 7a, pure poly is used to drive the WL. This has the best packing density but turns out to be the slowest. In Figure 7b, the WL is driven by poly,

and M3 is used to strap the WL every 16 bits. This provides the worst density but is the fastest. Figure 7c is similar to the second, except that M3 is used to strap the WL per 32 bits.

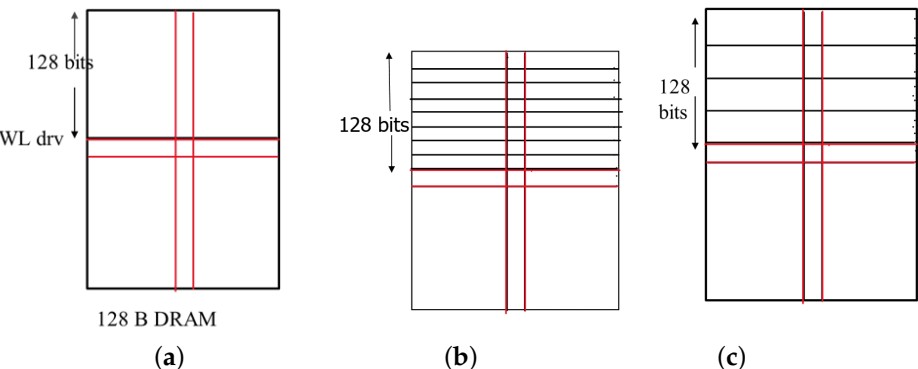

(a)  (b)  (c)

**Figure 7.** Baseline designs of DRAMs used for evaluating the performance of the hybrid poly-WSM interconnect: (**a**) Pure poly without any M3 strapping; (**b**) Pure poly with M3 strapping every 16 bits; and (**c**) Pure poly with M3 strapping every 32 bits.

We also evaluated the same designs using 512 B DRAMs, also by replacing poly routing with the poly-WSM hybrid interconnect.

*(b) D-FF:* D-FF is widely used in chip design in large quantities (tens of thousands). Therefore, compact and high-performance D-FF design is key to achieving area and energy efficiency. D-FFs also suffer from congestion due to vias and metals making it a perfect candidate for hybrid poly-WSM exploration.

**Area vs. performance trade-off for hybrid poly-WSM interconnect:** We use hybrid poly-WSM to route the WLs and compare the performance for all three cases of DRAM/eDRAM. We note that the highest improvement in performance is obtained when we compare the delays for a regular poly interconnect with a hybrid poly-WSM interconnect without strapping (Table 2). The delay becomes 1/3 of that for the usual poly interconnect. When M3 is used to strap the wordline every 16 bits, the improvement is approximately 12%. The performance gain is modest (approximately 7–8%) when the strapping is performed every 32 bits.

For a D-FF, we note from Figure 8a that M1 vias limit the footprint. The hybrid interconnects (Figure 8b) are used to route some of the signals which in turn eliminate the M1 and vias and reduces the area by 15.6%.

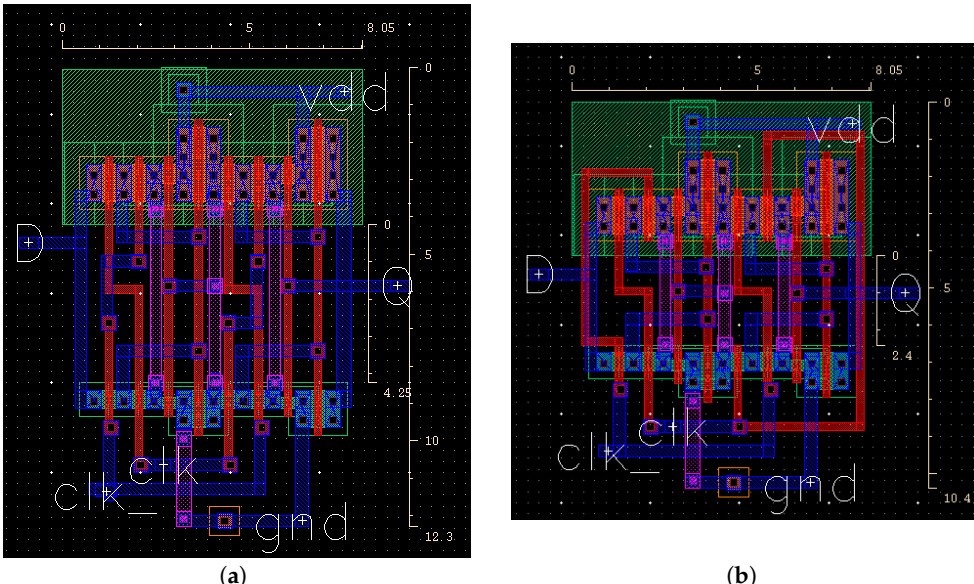

(a)  (b)

**Figure 8.** (**a**) Standard D flipflop and (**b**) D flipflop using hybrid poly-WSM interconnect.

**Table 2.** DRAM/eDRAM Performance.

| Design | Delay | % Benefit |
|---|---|---|
| Poly-256B<br>Poly-WSM-256B | 530.4 ps<br>177 ps | 66.6% |
| Polystrapping-256B-(WL1-WL16)<br>Poly+WSM-strapping-256B-(WL1-WL16) | 32.7 ps<br>29.9 ps | 8.56% |
| Polystrapping-256B-(WL1-WL32)<br>Poly+WSM-strapping-256B-(WL1-WL32) | 43.8 ps<br>32.6 ps | 25.57% |
| Poly-512B<br>Poly-WSM-512B | 1289 ps<br>442 ps | 65.7% |
| Polystrapping-512B-(WL1-WL16)<br>Poly+WSM-strapping-512B-(WL1-WL16) | 126.9 ps<br>118 ps | 7.01% |
| Polystrapping-512B-(WL1-WL32)<br>Poly+WSM-strapping-512B-(WL1-WL32) | 136.9 ps<br>120 ps | 12.34% |

However, it has some impact on the performance: in this case, the clk-Q and D-Q delay is degraded. The delay performance is, however, better for a hybrid poly design than for a pure poly-based design and can be further optimized (Table 3). The poly-WSM hybrid interconnects can be used in off-critical paths to avoid performance issues while simultaneously reaping the benefits of area reduction.

**Pathfinding for a higher-level interconnect:** To evaluate the performance of the NbAs-based interconnect, we first varied the length of the ribbon from 1 mm to 10 mm (width: 0.1 μm) and measured the variation in propagation delay and resistance. As expected, the resistance (Figure 9a) and propagation delay (Figure 10b) increases with the nanoribbon length. Furthermore, NbAs is a better conductor than Cu as is evident from the plots, providing lower delay and resistance. Figure 10a shows that the maximum possible delay improvement for NbAs ribbon for a 22 nm process technology is 20.9%.

**Table 3.** Performance comparison between different layouts of D Flipflop.

|  | Regular Layout | Using Poly (1 μm) | Using Poly + WSM |
|---|---|---|---|
| D to Q | 0.1 ns | 0.35 ns | 0.21 ns |
| Clk to Q | 0.05 ns | 0.28 ns | 0.12 ns |

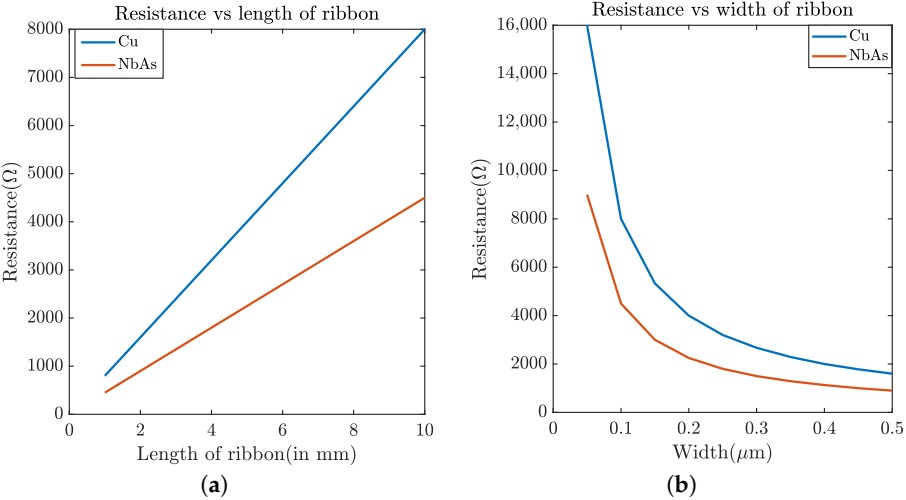

**Figure 9.** Resistance of (**a**) 100 nm-wide nanoribbon for Cu and NbAs with respect to length. (**b**) 10 mm-long nanoribbon with respect to width.

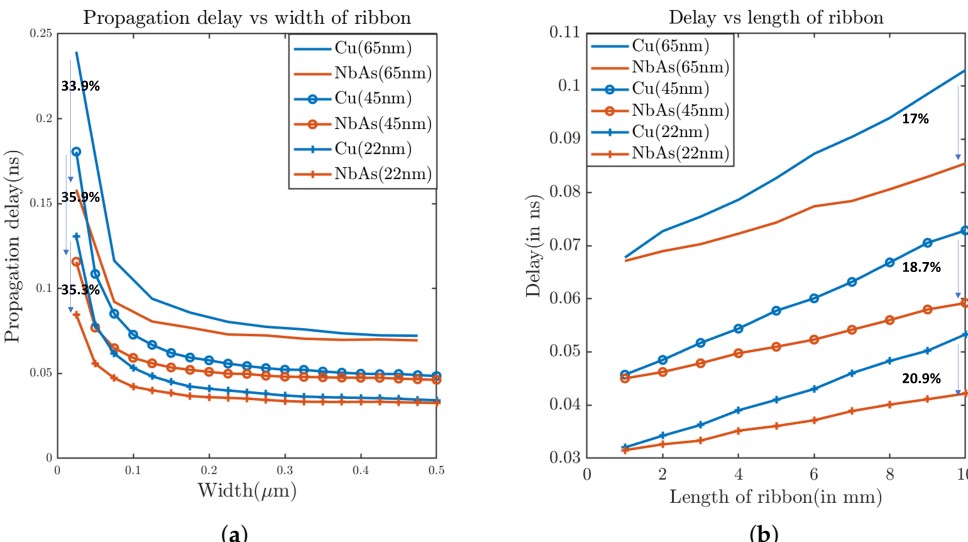

(**a**)          (**b**)

**Figure 10.** Delay comparison between Cu and NbAs for 65 nm, 45 nm, and 22 nm process technologies with respect to (**a**) width of the nanoribbon and (**b**) length of the nanoribbon.

We also varied the width of the ribbon from 0.025 μm to 0.5 μm (length: 10mm) and observed the delay and resistance. Increasing the width increased the cross-sectional area of the ribbon (a thickness of 200 nm) and thus the resistance dropped (Figure 9b), subsequently reducing the propagation delay (Figure 10a). We observed that the maximum delay improvement provided by NbAs is 35.88% for the 45 nm process technology when the width of the ribbon is 0.025 μm. Figure 14a shows the maximum delay improvement for various process technologies.

A general guideline for the slew time is to keep it below 100 ps to mitigate potential signal integrity issues. For this study, the ratio of widths of the transmitter inverter and the receiver inverter is varied from 1:4 to 1:36. From Figure 11, we can note that NbAs outperforms Cu in terms of slew time. As the size of the receiver inverter increases, the propagation delay also increases. Figure 12 shows that the percentage delay improvement for NbAs with respect to Cu increases as the width ratio of the transmitter and the receiver inverter increases. When the width of the ribbon is varied with a fixed receiver inverter size (Figure 13), the slew time falls since the resistance falls with an increase in width.

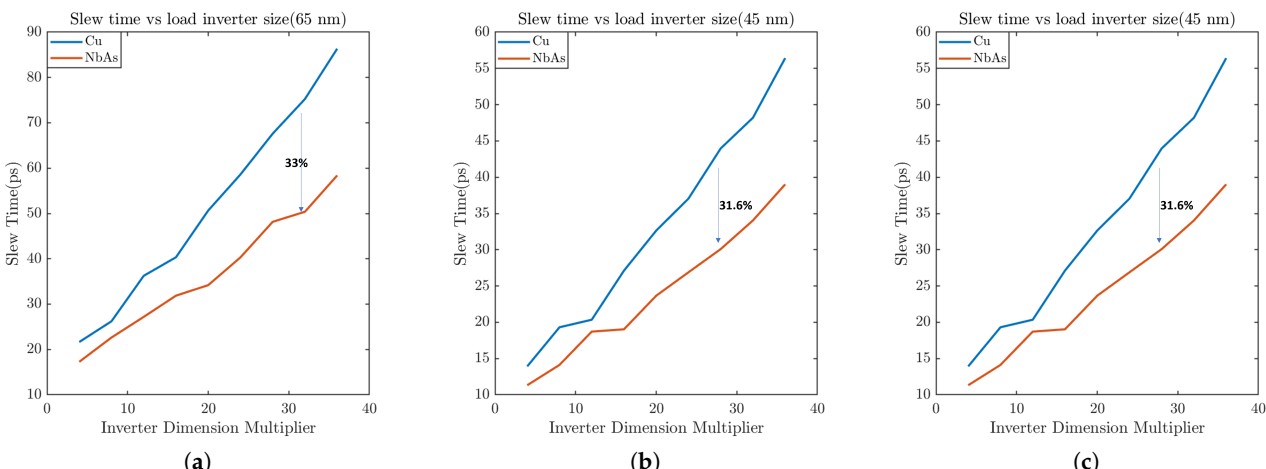

(**a**)          (**b**)          (**c**)

**Figure 11.** Slew rate comparison between Cu and NbAs with respect to the width of the receiver inverter for (**a**) 65 nm, (**b**) 45 nm, and (**c**) 22 nm process technologies.

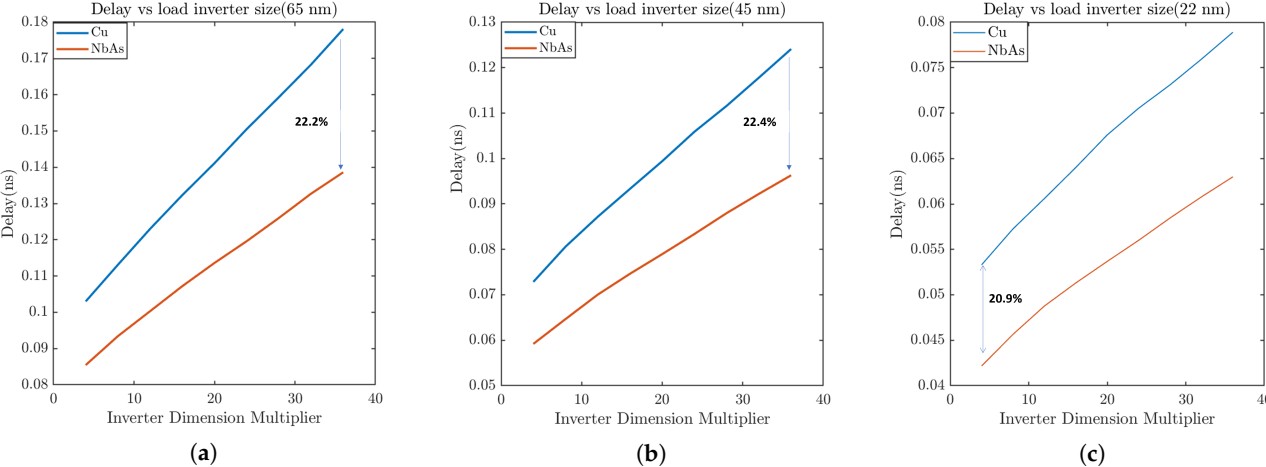

**Figure 12.** Propagation delay comparison between Cu and NbAs with respect to the width of the load inverter for (**a**) 65 nm, (**b**) 45 nm, and (**c**) 22 nm process technologies.

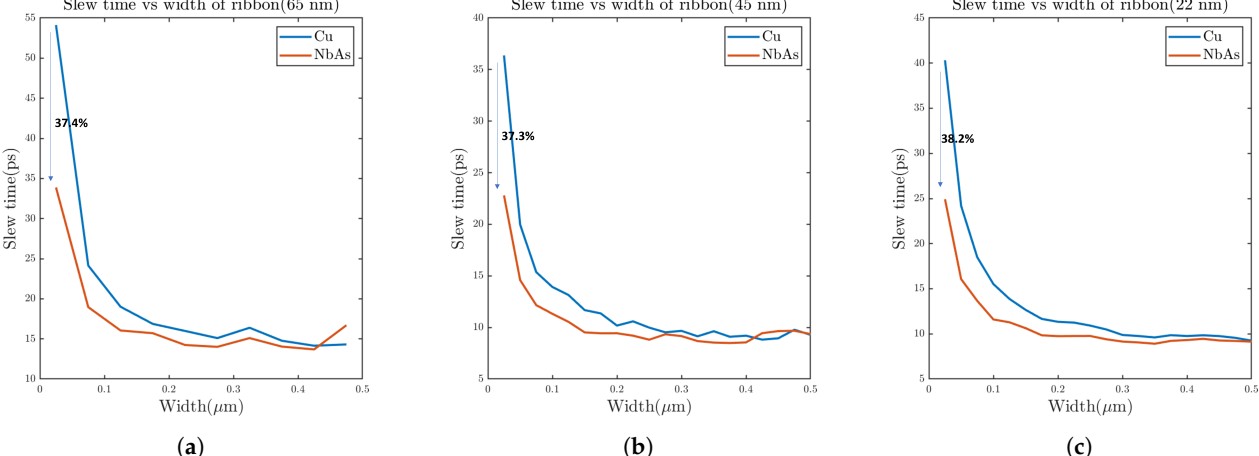

**Figure 13.** Slew rate comparison between Cu and NbAs with respect to the width of the nanoribbon for (**a**) 65 nm, (**b**) 45 nm, and (**c**) 22 nm process technologies.

**System-level performance evaluation:** Figure 10a shows that for the 22 nm process technology (at length and width of the wire fixed at 10 mm and 25 nm, respectively), the propagation delay for NbAs is 35.28% lower when compared to Cu. Table 4 lists the interconnect properties corresponding to the propagation delay improvement. As discussed earlier, although the total cache latency is dependent on both the interconnect and cache access latency, interconnect latency is the major bottleneck because access latencies are very fast, i.e., typically 1 cycle [18]. Hence, taking the delay improvement in mind, we reduced both the L1 and L2 cache latencies, keeping all the other parameters constant to analyze the performance improvement provided by NbAs interconnects. From Figure 14b, we can see that, as expected, with a decrease in cache latency (in the case of NbAs), the IPC increases. This is because lower cache latency indicates that the cache requires fewer clock cycles to perform the required operations, resulting in a reduction in the total clock cycles used by the overall system and thus increasing the number of instructions executed per cycle, i.e., IPC. The IPC improvement ranges from 12.7% in the case of canneal to as high as 23.8% in the case of streamcluster. The average improvement in IPC provided by NbAs across all benchmarks is 18.56%. The actual improvement amount would vary depending on the workload. Intuitively, we understand that workloads that require frequent memory access would benefit more.

We further considered a more realistic global interconnect scenario where the width and length of the interconnect wire are 100 nm and 10 mm, respectively. The corresponding propagation delay improvement for the 22 nm node is found to be 20.9% (Figure 10b).

**Table 4.** The interconnect properties that correspond to the maximum propagation delay improvement provided by NbAs over Cu. We also included pure poly resistivity, which was used in the low-level interconnect evaluation.

| Parameter | Value |
|:---:|:---:|
| Length | 10 mm |
| Width | 0.025 μm |
| Aspect ratio | 8:1 |
| Resistivity (Cu) | 1.6 μΩ cm |
| Resistivity (NbAs) | 0.9 μΩ cm |
| Resistivity (Poly) | 2 μΩ cm |

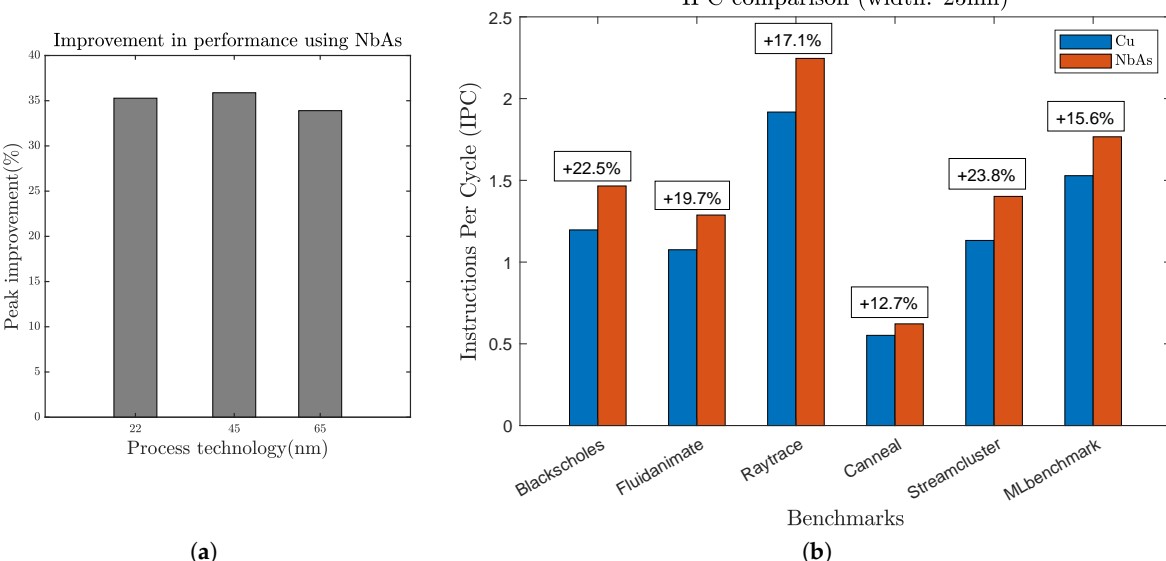

(**a**)   (**b**)

**Figure 14.** (**a**) Maximum propagation delay improvement corresponding to different process technologies. (**b**) Figure demonstrating the IPC improvement corresponding to a 35% reduction (22 nm technology) in propagation delay provided by NbAs (over Cu) on various benchmarks.

We reran our gem5 simulations considering this improvement and found the IPC to improve by up to 15.7% in the case of the blackscholes benchmark. The average IPC improvement across all the benchmarks was 13.67%.

We also analyzed the total execution time (ET) improvement provided by NbAs interconnects. We showed this on top of IPC improvement because the total execution time is a widely recognized performance metric and is known as the "Iron Law of Performance" [19]. The total execution time for a single thread program with $N$ instructions can be calculated as follows:

$$ET = N \cdot CPI \cdot \frac{1}{f}$$

where $CPI$ refers to the clock cycles per instruction and $f$ refers to the cpu clock frequency.

The Iron Law of Performance is useful because the terms correspond to the sources of performance. The quantity of instructions $N$ is determined by the instruction-set architecture (ISA) and the compiler; $CPI$ is determined by the microarchitecture and the circuit-level implementation, and $f$ is determined by the circuit-level implementation

and technology. Improving one of these three performance sources improves overall performance [20]. Figure 15a shows the execution time improvement (in %) provided by NbAs over Cu. As expected, similarly to the IPC improvement, the improvement varies from 11% provided by the canneal benchmark to 19% provided by the streamcluster benchmark. The average improvement in execution time across all benchmarks is 15.95%. Table 5 displays the host memory usage and L2 cache miss rate for the various benchmarks when run on the gem5 simulator to provide a better understanding of the workload of these benchmarks.

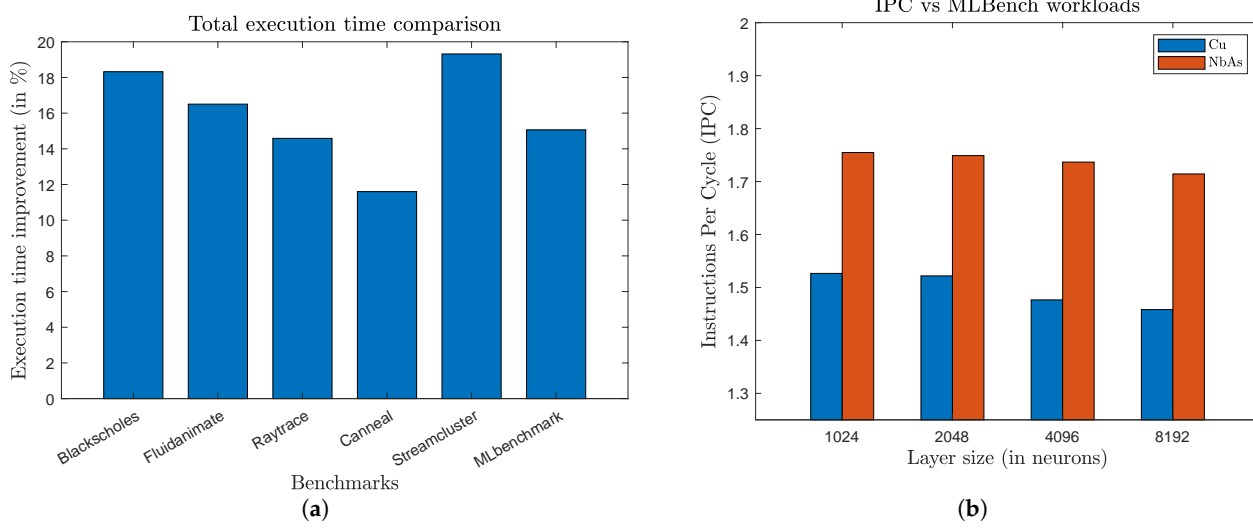

(a)                                    (b)

**Figure 15.** (**a**) Total execution time improvement provided by NbAs on different benchmarks. (**b**) Figure demonstrating the change in IPC for different ML benchmark workload sizes (i.e., increasing neurons in each hidden layer of the MLP).

**Table 5.** Memory usage and cache miss rate of different parsec benchmarks used in this study.

| Benchmark | Memory Usage (in kB) | L2 Miss Rate |
|-----------|----------------------|--------------|
| Blackscholes | 698 | 0.20 |
| Fluidanimate | 735 | 0.63 |
| Raytrace | 820 | 0.77 |
| Canneal | 741 | 0.42 |
| Streamcluster | 730 | 0.88 |

We also tested the ML benchmark with different workloads (model sizes), to see the impact it has on IPC and the total memory used. Specifically, we varied the layer sizes of the neural network model to create different-sized workloads. Table 6 shows the memory used by the different benchmark workloads (models), with hidden layer sizes ranging from 512 neurons to 8196 neurons on the gem5 simulator. As expected, the host memory used by the benchmark increases as we increase the model size since the number of parameters increases. Figure 15b shows the IPC for Cu and NbAs on the ML benchmark with different workloads. From the figure, we can understand that there is very minimal change in IPC with the increase in neural network model size. However, the IPC improvement increases by 2% as we increase the model size from 1024 neurons to 8192 neurons in hidden layers.

**Table 6.** Memory usage of different sized ML models (workloads) used for benchmarking.

| NN Layer Sizes (in Neurons) | Memory Usage (in kB) |
| --- | --- |
| 512 | 668 |
| 1024 | 673 |
| 2048 | 687 |
| 4096 | 742 |
| 8192 | 951 |

## 4. Conclusions

With the continued shrinking of transistor dimensions, metal interconnects are now becoming the speed bottleneck in today's most advanced chips. In this paper, we showed that hybrid poly with WSM coating for local routing in circuit improves the performance of DRAM/eDRAM by up to 3X and reduces the area of D-FF by 15.6%. We also demonstrate how NbAs outperforms Cu when used as interconnects at the nanoscale with respect to conductivity, slew time, propagation delay, etc. We show that using NbAs could reduce the propagation delay and slew time by up to 35.88% and 38%, respectively. We also demonstrated the system-level performance improvement it could provide by running extensive simulations using the gem5 simulator. Our analysis showed that using NbAs global interconnects improves the cache performance, and as a result, improves the overall system performance, i.e., the IPC and total execution time by up to 23.8% and 19%, respectively.

**Author Contributions:** Conceptualization, S.H. and S.G.; methodology, S.H. and S.G.; software, S.K., R.R. and S.U.; validation, S.G., S.H. and R.O.T.; formal analysis, S.K., R.R. and M.S.R.; investigation, S.H., S.G., S.E.M. and R.O.T.; resources, S.H. and S.G.; data curation, S.K., R.R., M.S.R. and S.U.; writing—original draft preparation, S.K., R.R., M.S.R. and S.H.; writing—review and editing, S.G., S.H., R.O.T. and S.E.M.; visualization, S.K., R.R. and M.S.R.; supervision, S.H., S.G. and R.O.T.; project administration, S.H., S.G. and R.O.T.; funding acquisition, S.H. and S.G. All authors have read and agreed to the published version of the manuscript.

**Funding:** This research was funded by Semiconductor Research Corporation (3011.001).

**Data Availability Statement:** Not applicable.

**Acknowledgments:** This work is supported by NSF (CNS-1722557, CCF-1718474, DGE-1723687, DGE-1821766, OIA-2040667, DGE-2113839, ECCS-2246564, and ECCS-1943895) and SRC LMD Task 3011.001.

**Conflicts of Interest:** The authors declare no conflict of interest.

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
