# Peer review of "Exploring Topological Semi-Metals for Interconnects"

_jlpea, doi:10.3390/jlpea13010016_

Round 1

Reviewer 1 Report

Satwik Kundu and co-workers study the effect of topological semimetals for interconnects on the the performance gain for DRAM/eDRAM. The authors propose interesting materials and method to developing next generation electronics. Overall, the main points, sensing performance and mechanism, are well stated with appropriate experiments and calculation, so I believe that this manuscript can be published after addressing some minor revision points.

1. The combination of various WSMs for high and low levels was an impressive approach. However, it seems that the paper's explanation of the effect at the low level is insufficient. How much does CoPt effect performance improvement when used at a low level?

2. Figure 4b shows a impressive change in resistivity of CoPt thin film. It would be great if the author could include the results of the grain size analysis by thickness.

Author Response

We appreciate your precious time in reviewing our paper and providing valuable comments. Please see the attachment for our responses.

Reviewer 2 Report

In this paper, the authors propose to use CoPt for local and NbAs for high-level interconnects. Interesting and promising results are shown for DRAM and system-level benchmarks, but the authors need to address and clarify the followings.

It’s better to have a table to list the interconnect properties (e.g. width, aspect ratio, resistance per unit length) for local and global interconnects used in DRAM and system-level benchmarks.

For system-level evaluation, it is very rare to use a narrow width of 25nm for a global long interconnect of 10mm. Can the authors consider more practical interconnect geometry (e.g. >100nm) for the case study?

Please use the log scale for figure 5 because it is difficult to see the difference between Cu and NbAs at small values.

For NbAs-based interconnect, fixed buffer distance with fixed strength were used. However, in practical design, optimal repeater insertion is used with optimal buffer distance and strength. Different materials may have different optimal design. Can the author consider that by sweeping buffer numbers and strengths to have a proper and fair comparison with Cu?

There are some grammatical issues with the paper. Please proofread more carefully. For example:

On page 9 line 252, there are two duplicated “the” in the sentence “we reduced the both the L1 and L2 cache latencies keeping all the other parameters constant to analyze the performance”.

Author Response

(The authors gave the same response as above.)

Round 2

Reviewer 2 Report

Thanks for the revision. In table 4, the aspect ratio of an interconnect is its height over width instead of length over width. It's better to show the Cu counterpart as well in the same table. Also, resistivity should have a unit of Ohm*m instead of Ohm.
